# Measurement report: Plume heights of the April 2021 La Soufrière eruptions from GOES-17 side views and GOES-16–MODIS stereo views

Ákos Horváth[1], James L. Carr[2], Dong L. Wu[3], Julia Bruckert[4], Gholam Ali Hoshyaripour[4], Stefan A. Buehler[1]

[1]Meteorological Institute, Universität Hamburg, Hamburg, Germany
[2]Carr Astronautics, Greenbelt, MD, USA
[3]NASA Goddard Space Flight Center, Greenbelt, MD, USA
[4]Institute of Meteorology and Climate Research, Karlsruhe Institute of Technology (KIT), Karlsruhe, Germany

*Correspondence to*: Ákos Horváth (akos.horvath@uni-hamburg.de, hfakos@gmail.com)

**Abstract.** We estimated geometric plume heights for the daytime eruptions of La Soufrière in April 2021 using visible red band geostationary side views and geostationary–polar orbiter stereo views. Most of the plumes either spread near the tropopause at 16–17km altitude or penetrated the stratosphere at 18–20km altitude. Overshooting tops reached heights up to 23km. These geometric heights were compared with radiometric heights corresponding to the coldest plume temperature, which usually represent ambiguous estimates within a wide range between a tropospheric and stratospheric height match. The tropospheric lower bound of the radiometric height range always underestimated the geometric height by a couple of kilometers, even for smaller plumes. For plumes near or above the tropopause, the midpoint or the stratospheric upper bound of the radiometric height range was in reasonable agreement with the geometric heights. The geometric overshooting top height, however, was always above the radiometric height range. We also found that geometric plume heights can be estimated from infrared band side views too, albeit with increased uncertainty compared to the visible red band. This opens the possibility for applying the side view method to night time eruptions.

## 1 Introduction

The La Soufrière stratovolcano (also known as Soufrière St. Vincent; 13.33ºN, 61.18ºW) on St. Vincent Island in the Lesser Antilles erupted on 9 April 2021, almost exactly 42 years to the day of its last major eruption in April 1979 (Fiske and Sigurdsson, 1982). The multi-day eruption was observed by the Advanced Baseline Imager (ABI) aboard Geostationary Operational Environmental Satellite-16 (GOES-16, GOES-East) and GOES-17 (GOES-West), providing full disk (FD) imagery at 10-minute frequency. The GOES-16 mesoscale sector (MESO2) was centered over the volcano at 09:00UTC on 10 April, providing 1-minute imagery of the plume in a 1000×1000 km$^2$ domain until 05:59UTC on 16 April. By tracking the emergence of cold bubbles near the volcano in animated infrared (IR) brightness temperature images, we counted 49 eruptions until 22 April, although it is noted that pinpointing the start and end of individual pulses is somewhat subjective.

The first eruption occurred at 12:40UTC on 9 April, followed by five more on that day. The 10th and 11th of April saw the most intense activity, with 22 and 9 eruptions, respectively. On the 12th, 13th, and 14th of April there were four, three, and two eruptions. Finally, the 16th, 18th, and 22nd of April had one eruption each. This series of eruptions released a significant amount of ash and $SO_2$ into the free troposphere, caused widespread ashfall on St. Vincent and neighboring islands including Barbados (165 km east), and prompted the evacuation of tens of thousands of people (Global Volcanism Program, 2021). The plumes mostly drifted east-northeast in the northern hemisphere and reached Taiwan 10 days after the initial eruption on 19 April (Babu et al., 2022).

The GOES-16 and GOES-17 view geometries for La Soufrière are plotted in Fig. 1. GOES-16, stationed at 75.2ºW, observes the volcano from the southwest (view azimuth of -133º) at a small view zenith angle (VZA) of 22.4º. GOES-17, stationed at 137.2ºW, observes the volcano almost exactly from the west (view azimuth of -93.5º) at a very large VZA of 84.9º, thanks to La Soufrière's location near the limb of the GOES-17 FD image. Such oblique observations allow plume height estimation by the recently introduced geometric side view technique (Horváth et al., 2021a, 2021b). Plume height can also be estimated by the traditional radiometric method of matching the minimum 11μm brightness temperature ($BT_{11}$, band 14) or 'dark pixel temperature' to a temperature profile (de Michele et al., 2019; Oppenheimer, 1998; Prata and Grant, 2001).

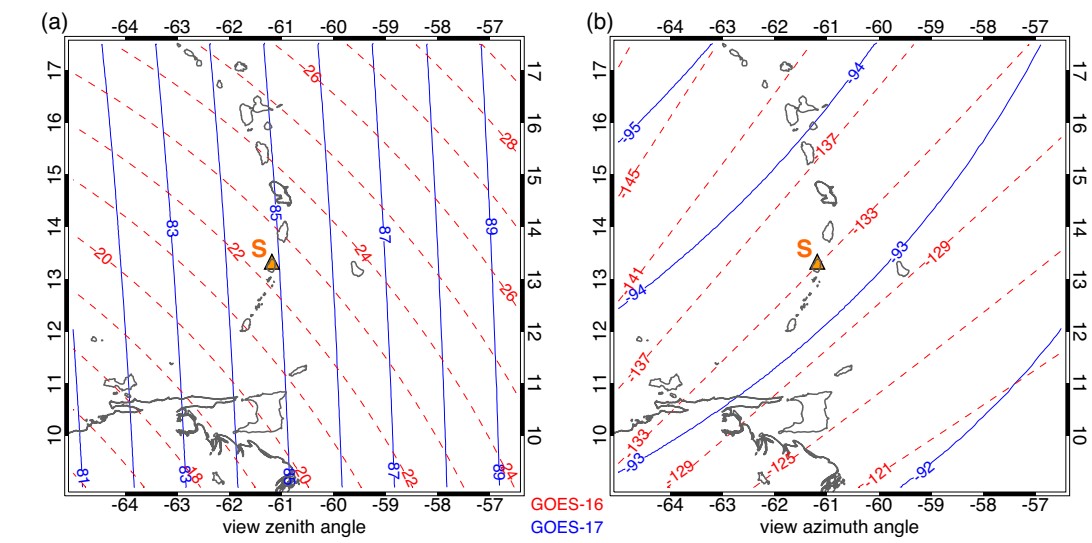

**Figure 1.** GOES-16 (dashed red) and GOES-17 (solid blue) **(a)** view zenith angle and **(b)** view azimuth angle for La Soufrière (orange triangle and letter 'S'). The negative view azimuth angle is measured counterclockwise from north.

In this measurement report, we derive daytime plume heights from 30 GOES-17 band 2 (red, 0.65μm) visible images that facilitate the side view technique. These geometric heights are compared with temperature-based heights corresponding to the GOES-16 dark pixel $BT_{11}$ of the plume. At the overpass times of the Terra and Aqua satellites, the results are also

validated with stereo heights retrieved by the automated "3D Winds" algorithm (Carr et al., 2019) using GOES-16 and Moderate Resolution Imaging Spectroradiometer (MODIS) visible red band images of the plume. A broad comparison with Cloud-Aerosol Lidar and Infrared Pathfinder Satellite Observation (CALIPSO) far-field lidar heights is also provided.

The report is organized as follows. In Section 2, we briefly describe the side view, temperature-based, and stereo height retrieval techniques and discuss the GOES FD and MESO2 observation timelines. In Section 3, the different height retrievals are demonstrated for seven specific eruption plumes that represent a range of explosivity and observing conditions. In Section 4, we characterize the biases of the temperature method using all 30 cases of side view height estimates and also compare our results with plume heights measured during La Soufrière's 1979 eruption. Section 5 concludes the report with a summary and outlook.

## 2 Height estimation methods

### 2.1 GOES-17 side views

The near-limb portion of geostationary imagery provides close-to-orthogonal side views of eruption plumes protruding from the Earth ellipsoid. Such oblique observations facilitate point estimates of near-field plume height by determining the angular extent of the eruption column between the known vent location and the plume top (Horváth et al., 2021a). The measurement principle is sketched in Fig. 2a. The apparent height $\hat{h}$ is the product of the column's angular extent $\delta$ as observed by the sensor at a VZA of $\theta$ and the known distance $D$ between the vent and the sensor. Height $\hat{h}$ is measured along axis $\hat{Z}$, which is perpendicular to the look vector connecting the sensor to the vent. Because $\delta \ll 1$, the apparent height $\hat{h}$ is foreshortened by a factor of $\sin(\theta)$ compared to the true height $h$ measured along the local vertical axis $Z$. Foreshortening is a trivial error in near-limb views with VZA>80º, for which the technique was originally devised.

Foreshortening, however, becomes more severe at smaller VZA, because a unit angular sampling distance (14μrad/pixel in the visible band) corresponds to a larger and larger true height differential. Thus, the isolines of true height get increasingly compressed with decreasing VZA, as demonstrated in Figs. 2b, 2c, and 2d. Panels 2b and 2d show the same La Soufrière eruption plume observed respectively by GOES-17 at VZA≈85º and GOES-16 at VZA≈22º (see also Sect. 3.4). In the GOES-17 side view, the plume top can be easily located between 16–17km.

Height estimation, however, is rather difficult in the more overhead GOES-16 view. A key step is to visually determine the plume point that lies directly above the vent along the local vertical. This is relatively straightforward in the GOES-17 side view, which shows a nearly vertical column with a well-defined tip. In contrast, GOES-16 mostly observes the spreading umbrella at the top of the eruption column. In this case, the center of the ellipse fitted to the umbrella might be used as the characteristic point for height estimation. However, ellipse fitting can be uncertain when the umbrella is amorphous, which can cause a large uncertainty in the plume height estimate due to the severe foreshortening at small VZA.

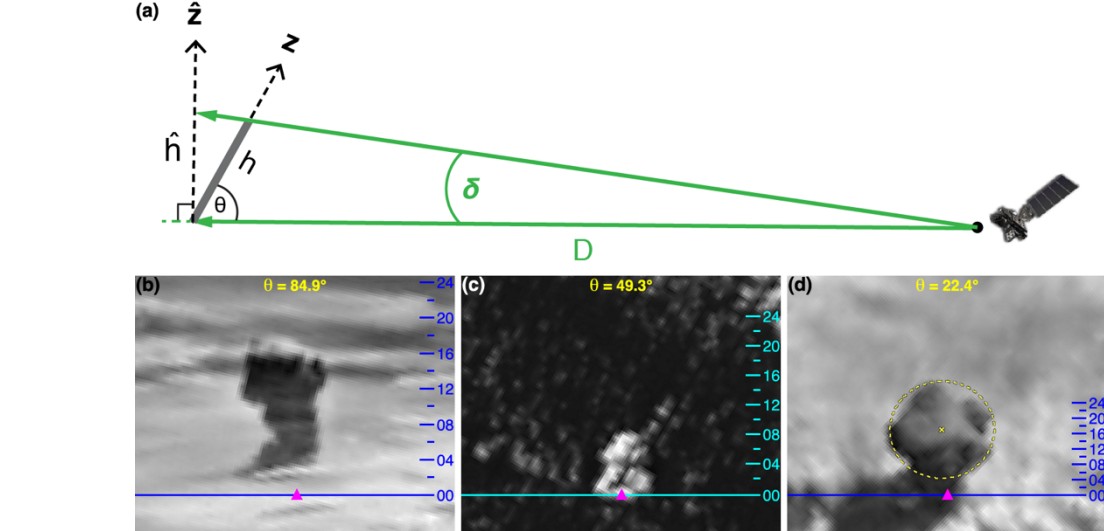

**Figure 2.** Side view measurement principle and increase of foreshortening at smaller view zenith angles. **(a)** The $Z$ axis is the local vertical and the $\hat{Z}$ axis is perpendicular to the sensor-to-volcano look vector. The true height and the apparent (foreshortened) height of the eruption column are $h$ and $\hat{h}$, respectively. The angular extent of the column is $\delta$, as measured from a distance $D$ and at a view zenith angle of $\theta$. Examples of eruption columns observed in channel 2 visible images (8× magnification) at decreasing $\theta$: **(b)** La Soufrière, 11 April 2021 at 13:30UTC by GOES-17, **(c)** Hunga Tonga-Hunga Ha'apai, 19 December 2021 at 20:30UTC by GOES-17, and **(d)** same as **(b)** but by GOES-16. The volcano is marked by the magenta triangle and the elevation markings indicate the true height in kilometres. In panel **(d)**, the yellow dashed line is a circle of 5.2km radius fitted to the umbrella.

Fig. 2c shows the GOES-17 view of a recent Hunga Tonga-Hunga Ha'apai eruption taken at an intermediate VZA of ~49º. The snapshot captures a rising column without a developed umbrella. Although the true height isolines are more densely packed than in the near-limb view of Fig. 2b, a column height of 7–8km can still be determined with relative ease.

The minimum VZA at which the side view technique is still useful depends on factors such as plume morphology, tilt, and wind speed and, thus, is a bit of a judgement call. In general, error in locating the plume top point directly above the vent causes larger errors at small VZA. Identifying this characteristic point is more uncertain at small VZA, when mostly the spreading umbrella is observed. The height error caused by wind-induced tilt or drift away from the local vertical is also larger in significantly foreshortened images. As an example, a 1-pixel error in the plume top location for the cases shown in Figs. 2b, 2c, and 2d introduces a height error of 502m, 660m, and 1312m, respectively.

Horváth et al. (2021a, 2021b) and the current study analyze eruptions that were imaged by GOES-17 at VZA>80º. With a relaxed constraint on VZA, the widened limb swaths of the operational geostationary satellites include many more volcanoes that could potentially be monitored with the side view technique. For illustration, Fig. 3 maps the locations of volcanoes that are observed at VZA>60º and which erupted in the past 100 years (historic eruption data were obtained from the Holocene

Volcano List of the Global Volcanism Program, 2013). The VZA>60º threshold is somewhat arbitrary, but in our experience height retrievals are still feasible at these angles. As shown, most of the major volcanic regions, including the Pacific Ring of Fire, are observed under relatively favorable (i.e., oblique) conditions by at least one satellite. Note that there are overlaps between the limb swaths and several regions are even imaged from opposite azimuths: Peru-Chile arc (from west by GOES-17 and from east by Meteosat-11), Iceland (from west by GOES-16 and from east by Meteosat-9), Kamchatka-Kuril arc and Papua New Guinea (from west by Feng-Yun-4A and Electro-L N3 and from east by GOES-17). Multiple independent retrievals would allow quality control by consistency checks and could provide more accurate height estimates for tilted plumes by averaging, because tilt errors are of opposite sign for opposite view azimuths.

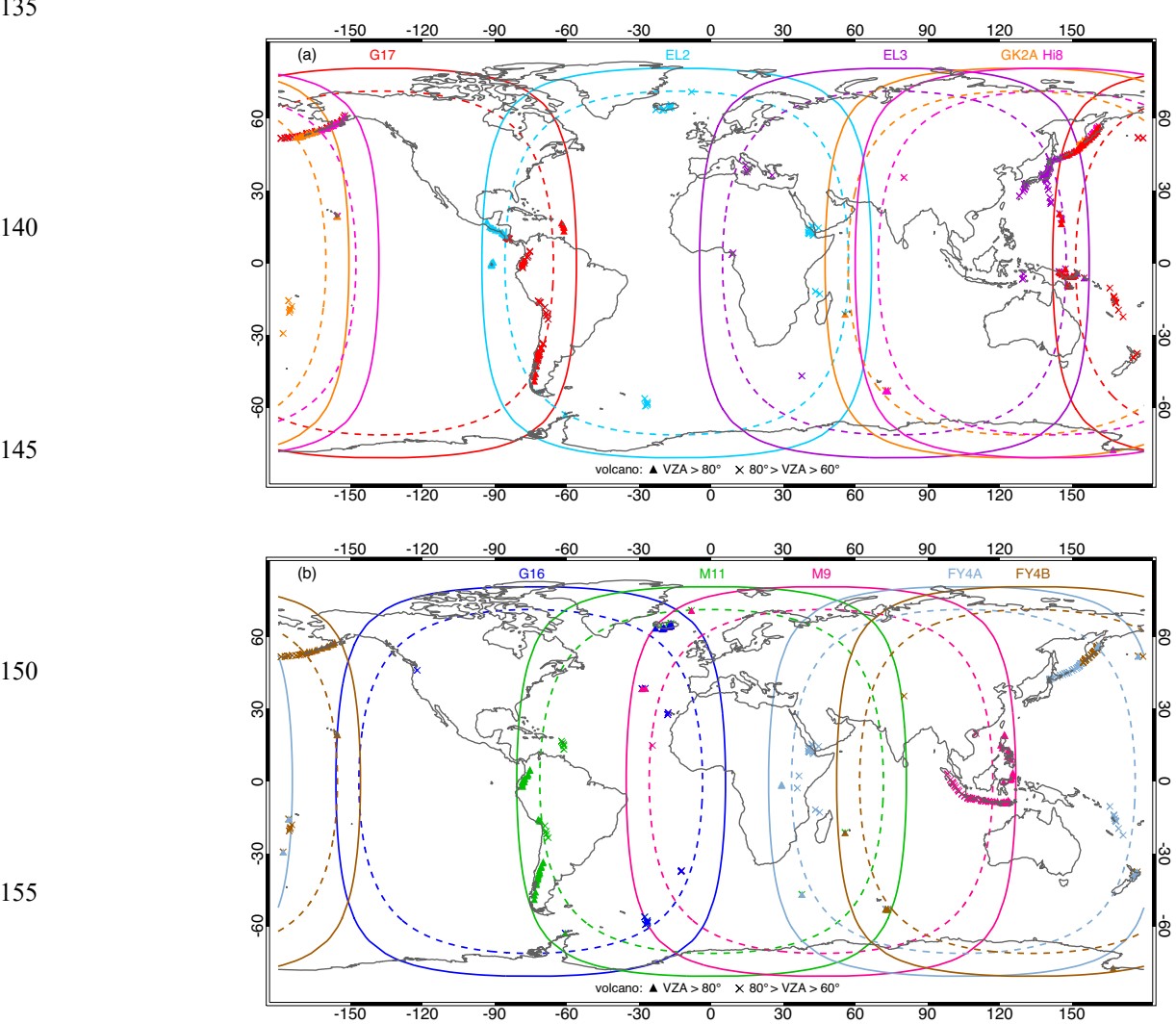

**Figure 3.** Limb areas of geostationary satellites between a VZA of 80º (dashed line) and 90º (solid line): **(a)** GOES-17 (G17, 137.2ºW), Electro-L N2 (EL2, 14.5ºW), Electro-L N3 (EL3, 76ºE), GEO-KOMPSAT-2A (GK2A, 128.2ºE), Himawari-8

(Hi8, 140.7ºE) and **(b)** GOES-16 (G16, 75.2ºW), Meteosat-11 (M11, 0º), Meteosat-9 (M9, 45.5ºE), Feng-Yun-4A (FY4A, 104.7ºE), Feng-Yun-4B (FY4B, 133ºE). INSAT-3D (82ºE) is not shown; its coverage is similar to that of EL3. Triangles indicate volcanoes that erupted within the limb areas in the past 100 years. Similarly, crosses indicate volcanoes imaged under a relaxed constraint of 80º>VZA>60º.

The technique is best suited to daytime visible red band images, which offer the highest horizontal resolution of 500m at the subsatellite point (Kalluri et al., 2018) and a vertically projected instantaneous field of view (or near-limb vertical resolution) which is only slightly coarserthan that. Besides GOES-16 and GOES-17, GEO-KOMPSAT-2A, Himawari-8, and Feng-Yun-4A/4B also carry a 500m visible band. The rest of the current fleet of geostationary satellites have visible bands with a resolution of 1–3km, but the next generation imagers will all have sub-km resolution channels.

The validation by Horváth et al. (2021b), which was limited to daytime cases with VZA>80º, found a typical height uncertainty of ±500m (or ±1 visible pixel) for near-vertical eruption columns. Although the current study also focuses on the analysis of visible images, we show that large plumes that reach the upper troposphere or lower stratosphere can be identified in near-limb IR images too. A similar ±1 IR pixel uncertainty in the measured vertical extent of a column corresponds to a ±2km height uncertainty due to the 4× coarser resolution of these bands. Such uncertainty can still be acceptable for nighttime height estimation, considering that radiometric methods have a typical uncertainty of 3–4km for high-level plumes (Thomas and Siddans, 2019).2.2 GOES-16 brightness temperaturesPlume height is also estimated with the traditional single-channel "temperature method", which matches the dark pixel $BT_{11}$ to the ERA5 (Hersbach et al., 2020) temperature profile. To avoid the limb cooling effects in GOES-17 data, we instead used the GOES-16 $BT_{11}$ obtained under small VZAs. Although these height estimates are subject to a number of potential errors (thermal disequilibrium, semitransparency, or uncertain chemical composition of the plume, temperature inversions), the temperature method is still an indispensable and oft-used tool thanks to its simplicity and the availability of IR radiometer channels aboard most meteorological satellites.

Figure 4 demonstrates the commonly arising problem of nonunique solutions in case of an inversion. Here we plotted the envelope of the night-time and daytime temperature profiles as well as the daytime-mean profile for 9–14 April. The atmospheric temperature structure within the eruption height range (< 24km) varied little during this period and was characterized by a strong inversion at the cold point tropopause located near 193.7K and 16.6 km. For this profile, plume temperatures colder than ~220K correspond to two height solutions: a tropospheric (minimum) one and a stratospheric (maximum) one. For example, for $BT_{11} = 210K$ the minimum plume height is $H_{P,min} = 13.5$km and the maximum is $H_{P,max} = 21.3$km. Because the tropospheric and stratospheric lapse rates are of opposite sign but comparable magnitude for a tropical temperature profile (-5.3K/km and +3.5K/km, respectively), the average of these two solutions, $H_{P,mean} = 17.4$km, gives a height near (slightly above) the tropopause. As we will show in Sect. 4.1, this midpoint height is the best match to the geometric height for a certain plume temperature range.

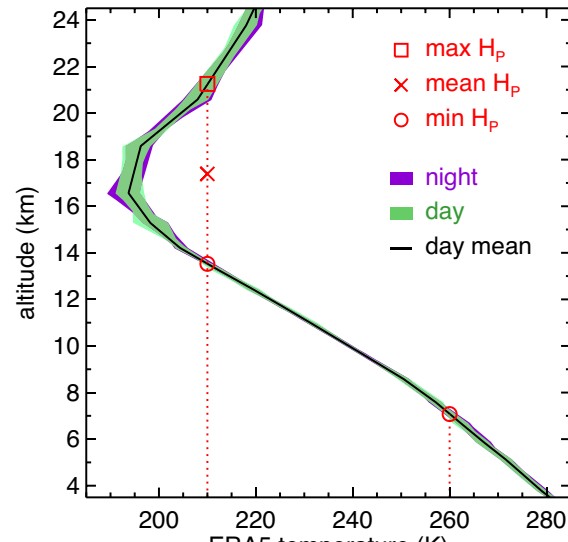

**Figure 4.** The envelope of night-time (magenta shading) and daytime (green shading) ERA5 temperature profiles and the daytime-mean temperature profile (black line) for La Soufrière over the main eruptive period of 9–14 April 2021. As a demonstration of the temperature method, the plume heights $H_P$ corresponding to a dark pixel temperature of 210K (two solutions) and 260K (single solution) are also marked.

In contrast, plume temperatures warmer than ~220K have a single tropospheric height match (for consistency still termed "minimum height"). For example, for $BT_{11} = 260$K the matching height is $H_{P,min} = 7.1$km.

Figure 4 also suggests that for stratospheric plumes, especially in the tropics, the maximum $BT_{11}$ near the center of the plume could be a better choice for radiometric height estimation. This was dramatically demonstrated by stereo height retrievals for the recent Hunga Tonga-Hunga Ha'apai eruption (Carr et al., 2022). This topic, however, is beyond the scope of the current study. Here, we use the minimum $BT_{11}$, as is customary, but consider all three possible radiometric heights (min, max, mean) for colder plumes and investigate which one is closest to the geometric height estimate.

## 2.3 GOES-16–MODIS stereo views

We also derive stereo heights for the Terra and Aqua overpasses on 10 April, by combining GOES-16 and MODIS red band images of the plume—the GOES–GOES combination could not be used due to the impossibility of template matching from low VZA to high VZA. The "3D Winds" algorithm applied here was developed for tracking wind tracers from multiple satellites; the version for a geostationary–polar orbiter pair is described in Carr et al. (2019). The technique retrieves both the height and the horizontal motion of a volcanic plume and has already been applied to Himawari-8–MODIS observations of

the 2019 Raikoke eruption (Horváth et al., 2021b) and Himawari-8–GOES-17 observations of the 2022 Hunga Tonga-Hunga Ha'apai eruption (Carr et al., 2022).

The algorithm requires a triplet of consecutive geostationary FD images and a single MODIS granule, the former temporally bracketing the latter. Feature templates are taken from the central repetition of the geostationary triplet and matched to the other two repetitions 10 min before and after, providing the primary source of plume velocity information. The geostationary feature template is then matched to the MODIS granule, which is observed from a different perspective and thus provides the stereoscopic height information. The apparent shift in the pattern from each match, modeled pixel times, and satellite ephemerides feed the retrieval model to enable the simultaneous calculation of the horizontal advection vector and its geometric height.

## 2.4 ABI observation timelines

During the eruption, GOES-16 operated in the default scan Mode 6, providing FD imagery every 10 minutes. GOES-17, on the other hand, followed the 15-minute FD scan Mode 3 cooling timeline between 06:00–12:00UTC to mitigate the loop heat pipe anomaly (McCorkel et al., 2019), and the 10-minute FD scan Mode 6 the rest of the day. Between 09:00UTC on 10 April and 05:59UTC on 16 April the 1-minute GOES-16 MESO2 observations were also available.

The ABI images are tagged by the scan start time, which is included in the radiance filename. La Soufrière, however, is observed ~3.3 minutes and ~4.1 minutes after the scan start time in Mode 6 and Mode 3 FD, respectively (Carr et al., 2020). Considering the slight time differences between scan start times too, the GOES-16 MESO2 trails the GOES-17 Mode 3 FD by 4–5 minutes. In contrast, the non-simultaneity between the GOES-16 and GOES-17 Mode 6 FD observations is less than 30 seconds in the same 10-minute slot. Therefore, we paired a GOES-17 FD with a near-simultaneous GOES-16 FD when both were acquired in Mode 6. A Mode 3 GOES-17 FD, however, was instead paired with the GOES-16 MESO2 trailing it by 5 minutes, in order to minimize the time gap between the geometric and radiometric height estimates.

## 3 Eruption examples

For each case, we plot two consecutive (10- or 15-minute) GOES-17 scans, while plume development over a 1-hour period is shown in the Supplement Animations. For visual clarity, the visible images were magnified by a factor of 8 and were enhanced by the Contrast Limited Adaptive Histogram Equalization (CLAHE) plugin of the Fiji package (Schindelin et al., 2012). The GOES-17 images were additionally rotated counter-clockwise by the geodetic colatitude (thus, top is ~east and bottom is ~west). Fixed grid data were used without any reprojection. Reported plume heights are above mean sea level rather than above the vent (summit elevation 1220m).

### 3.1 10 April, 09:45–10:00UTC

This eruption started during twilight, when the sun was still below the horizon at the volcano's location. In the 09:45UTC FD image (Fig. 5a, Supplement Animation 1), GOES-17 observes the western side of the towering eruption column against the background of the atmosphere illuminated by the sun rising in the distant east. The long shadow of the plume is faintly discernible with the column's gable-like top reaching an altitude of 22km in the contrast-enhanced side view. The upper half of the column above ~12km is also identifiable in the 11μm GOES-17 image by reduced brightness temperatures (Fig. 5c).

The resolution of this channel is 4× coarser than that of the visible red channel, nevertheless, the center of the IR pixel marking the top of the plume is near ~22km. The lower half of the plume, however, does not show enough temperature contrast against the background $BT_{11}$, which generally is subject to increased cooling near the limb due to water vapor absorption.

The corresponding 09:50UTC GOES-16 MESO2 visible image, offering more of an overhead view, shows the 265  overshooting top (OT) ascending above the illuminated parts of lower umbrella layers that spread near the level of neutral buoyancy (Fig. 5e). The parallax between the volcano and the OT is 9.3km as indicated by the yellow arrow. If the OT is assumed to lie above the vent, its height can be estimated from the parallax simply as $h = 9.3\mathrm{km}/\tan(\theta = 22.4°) = 22.6\mathrm{km}$, where $\theta$ is the view zenith angle. This back-of-the-envelope height estimate is consistent with the GOES-17 side view estimate, considering that the small GOES-16 VZA results in a relatively large ±1.2km height 270  error for a ±1pixel error in the parallax. Bending by the wind can introduce further height error—there were 16 m/s westerly winds at the tropopause—, although the OT being located along the view azimuth direction suggests small bending for this strong plume.

The GOES-16 dark pixel $BT_{11}$ of 191.8K is found at the OT location (Fig. 5g). Because this temperature is close to the ERA5 cold point, it corresponds to a narrow radiometric height range of 16.4–17.2km near the tropopause. The measured 275  temperature is ~20K colder than the ambient temperature at the OT height of 22km (see Fig. 4) and, thus, it is more representative of the umbrella height.

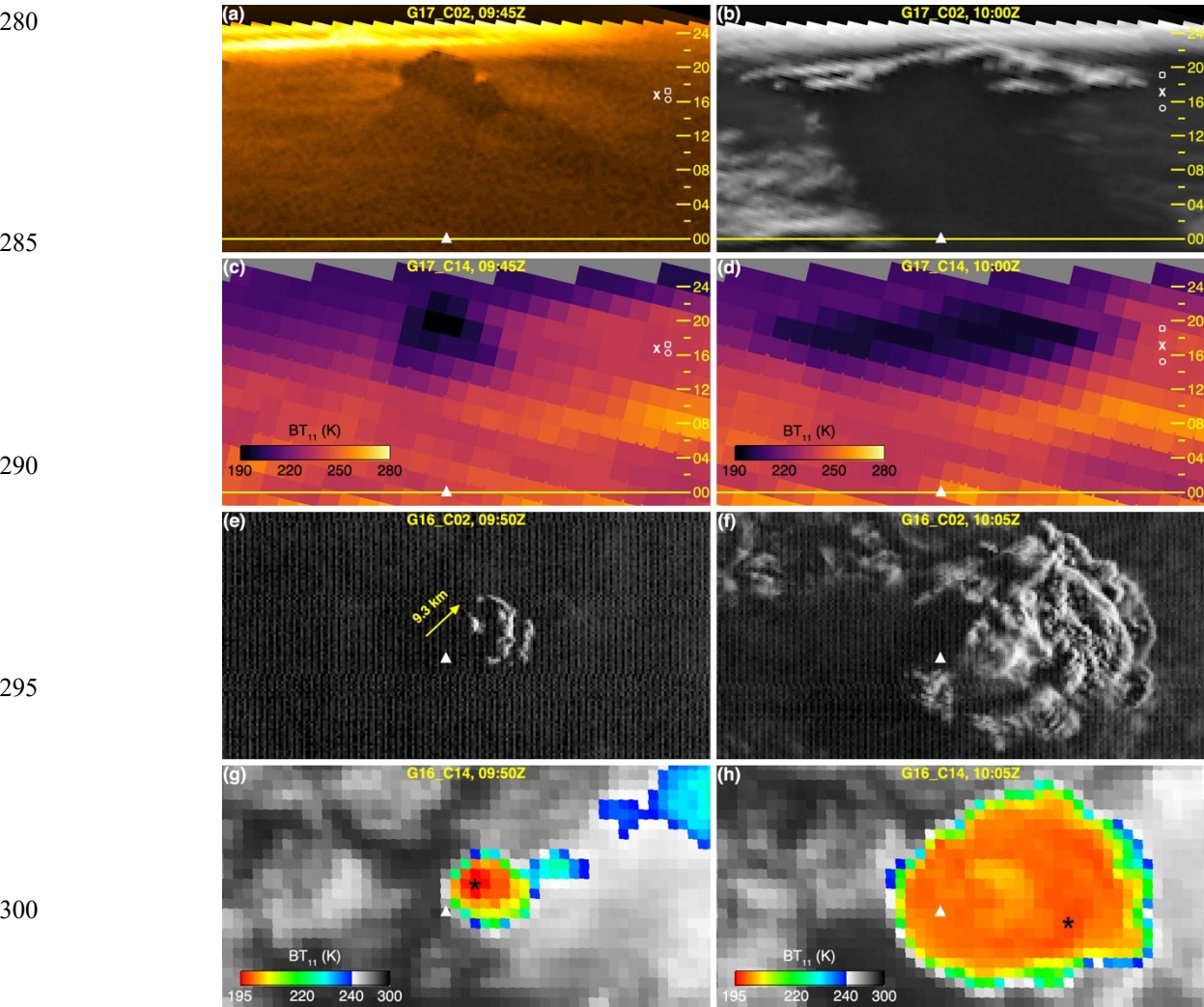

**Figure 5.** The eruption plume on 10 April 2021 at **(left column)** 09:45UTC and **(right column)** 10:00UTC in **(top to bottom)** GOES-17 channel 2, GOES-17 channel 14, GOES-16 channel 2, and GOES-16 channel 14 imagery. The GOES-16 images are from the trailing (+5 minutes) MESO2 scans. La Soufrière is marked by the white triangle and the image in panel **(a)** was pseudo-colored using the 'Orange Hot' palette. In the GOES-17 side views, the yellow line is the baseline, the elevation markings indicate height in kilometres and the white square, cross, and circle respectively depict the maximum, mean, and minimum plume height derived from the GOES-16 dark pixel temperature, whose location is marked by the black star in panels **(g)** and **(h)**. In panel **(e)**, the arrow indicates the ellipsoid-projected distance between the volcano and the overshooting top along the GOES-16 view azimuth of -133°.

By 10:00UTC the plume developed a large multi-layered umbrella (Fig. 5b and Fig. 5f). The dominant spreading level is at 18.0–18.5km with a collapsing OT at 21.0km according to the GOES-17 side view. The centerline of the umbrella can be located at ~18.0 km altitude in the GOES-17 IR image too (Fig. 5d). The GOES-16 plume-top $BT_{11}$ shows a cold ring

surrounding a central horseshoe-shaped warmer area (Fig. 5h), which is similar to the cloud-top IR patterns seen in severe deep convection (Setvák et al., 2013). The minimum $BT_{11}$ of 197.6K is located considerably downwind of the volcano. The upper (stratospheric) end of the corresponding radiometric height range of 15.3–19.1km agrees fairly well with the side view umbrella height estimate. Comparing the GOES-16 visible and IR images suggests that the central warm area is associated with the highest parts of the plume near the OT. The maximum temperature of this region is 203.8K, leading to an upper height solution of ~20 km, which is above the umbrella but still 1km below the side view OT height estimate.

### 3.1.1 Minimum plume height estimated from Earth's effective shadow height

As mentioned previously, when the first GOES-17 image was acquired (FD scan start time 09:45UTC, actual observation time 09:50UTC), the sun was still below the horizon at La Soufrière. The plume is discernible in the visible band images only because it rose above Earth's shadow and its top got illuminated. Calculating Earth's shadow height, thus, allows us to put an independent lower limit on plume height. The schematic of twilight observations of the plume is given in Fig. 6. The Earth's geometric shadow is defined by the point where the solar ray grazing the surface intersects the local vertical.

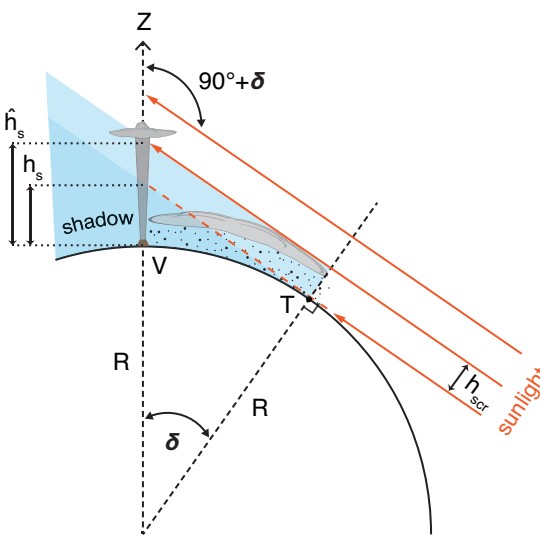

**Figure 6.** The geometry of an eruption column protruding through Earth's shadow at twilight. At a solar depression of $\delta$, a grazing ray tangent to the surface at point $T$ casts a geometric shadow of height $h_s$ at the volcano's location $V$. Grazing rays below the screening height $h_{scr}$ are strongly attenuated by air molecules, haze, and meteorological or volcanic clouds, raising the base of the scattering layer and thus increasing the effective shadow height $\hat{h}_s$. Here the spherical Earth's radius is $R$ and the apparent decrease in $\delta$ due to atmospheric refraction is omitted.

Here, 'geometric' refers to the shadow that Earth would cast if it had no atmosphere. For a spherical Earth of radius $R$ and an unrefracted solar depression angle of $\delta$, the geometric shadow height $h_s$ is

$$h_s = R(\sec \delta - 1). \tag{1}$$

The atmosphere introduces two opposing effects. First, refraction decreases the apparent solar depression by an angle $\omega$; this in itself reduces the shadow height. Second, below the so-called screening height $h_{scr}$, the atmosphere is nearly opaque to solar grazing rays due to strong attenuation through the long air path by molecules, haze, and potentially clouds. The screening height, which decreases with increasing wavelength, effectively raises the base of the scattering layer and thus the shadow height. With these two effects accounted for, Earth's effective shadow height can be written as

$$\hat{h}_s = (R + h_{scr}) \sec(\delta - \omega) - R. \tag{2}$$

Of the two effects, atmospheric screening is the easier to handle. Twilight photometry of aerosols and noctilucent clouds established that $h_{scr} = 7 \pm 1 km$ is a reasonable range for the red band screening height in typical cloud-free conditions

(Kumari et al., 2008; Taylor et al., 1984). In our case, however, the atmosphere between the volcano and the tangent point $T$ (located ~275km from the volcano along a solar azimuth of 81º) was covered by a thick ash cloud from prior eruptions as well as cirrus clouds. The $BT_{11}$ near the tangent point varied between 210–220K, suggesting a screening height of $h_{scr} = 12$–$13 km$. As we show later, the side view and stereo retrievals also indicated cirrus at 12–13km altitude.

The twilight refraction effect, however, can only be roughly estimated. It is hopeless to predict refraction accurately near

and below the horizon, because it depends on the lapse rate in the boundary layer, which is simply too variable due to weather (Young, 2004). Sunrise and sunset observations revealed that a reasonable range for the variation of the horizontal refraction angle for an unknown site is ~0.64º around the value predicted for standard conditions (Schaefer and Liller, 1990). In our work, the grazing ray refraction at the surface $\omega_0$ was interpolated to the encountered solar depression angles from the standard values given in Garfinkel (1967), resulting in a typical range of $\omega_0 \pm 0.32° \approx 0.7°$–$1.4°$. These surface refraction

angles were then pressure-scaled to the screening height of 12–13km (or 15–20% of the surface pressure), leading to a final refraction angle range of $\omega \approx 0.11°$–$0.27°$. For such a large screening height, which is the dominant factor in our case, the refraction correction amounts to a relatively small, at most ~1.0km reduction in shadow height.

As shown in Fig. 7, umbrella layer 1 ($U_1$) first became visible in the 09:48UTC MESO2 image. In the next three minutes, a second umbrella layer ($U_2$) and the OT emerged, then expanded and moved eastward. Earth's effective shadow height,

calculated from Eq. (2) using the indicated solar depression angle and the atmospheric screening and refraction corrections discussed above, decreased by 0.8–0.9km per minute.

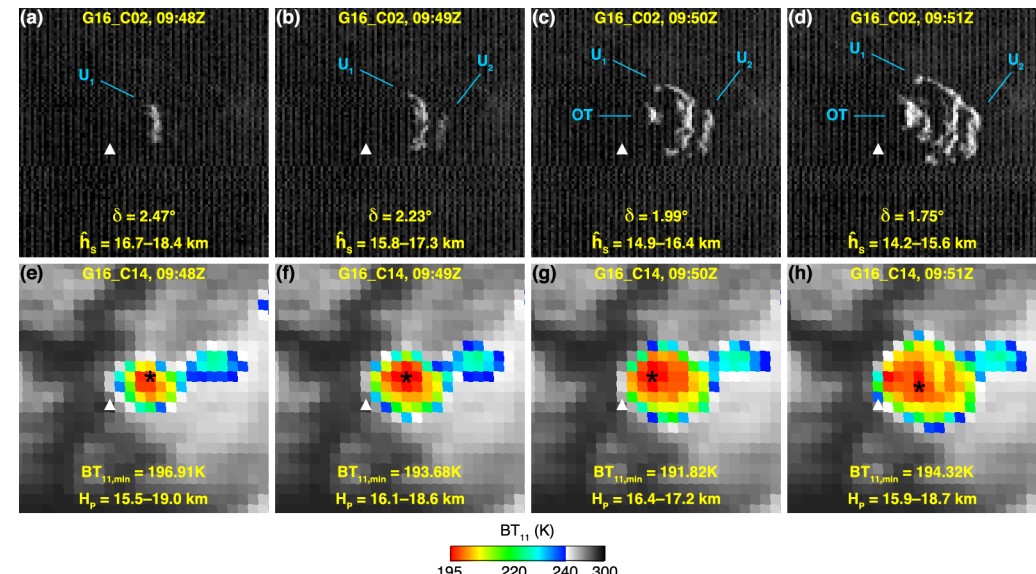

**Figure 7.** Minute-scale evolution of the eruption plume on 10 April 2021 between **(left to right)** 09:48–09:51UTC in GOES-16 MESO2 imagery: **(top row)** channel 2 and **(bottom row)** channel 14. The labelled plume features are the umbrella layer 1 ($U_1$), umbrella layer 2 ($U_2$), and overshooting top (OT). The unrefracted solar depression angle $\delta$, the estimated effective shadow height $\hat{h}_s$, the dark pixel temperature $BT_{11,min}$ and its location (black star), and the corresponding min–max radiometric range of plume height $H_P$ are also indicated.

The unrefracted solar depression was computed with the Solar Geometry Calculator of the National Oceanic and Atmospheric Administration (NOAA) Global Monitoring Laboratory (https://gml.noaa.gov/grad/antuv/SolarCalc.jsp). Plume height must increase from east to west, i.e. $H_{OT} > H_{U_1} > H_{U_2}$, because the eastern side of each of these layers gets illuminated by the rising sun (i.e. there is no obscuration by the adjacent layer to the east). Using the lower end of the shadow height range, we can conservatively estimate that $H_{U_2} > 15.8$km and $H_{OT} > H_{U_1} > 16.7$km, that is, the OT reached at least the tropopause. A less conservative estimate based on the upper end of the shadow height range suggests a minimum OT height of 18.4km.

### 3.2 10 April, 16:20–16:30UTC

This was one of the two most intense daytime eruptions. At 16:20UTC, the rising column with a pileus on top is captured at an altitude of 10.5–11.0km in the GOES-17 side view (Fig. 8a, Supplement Animation 2). The GOES-16 minimum $BT_{11}$ of 245.0K corresponds to a single underestimated height solution of 9.3km. At 16:30UTC, the plume features an OT at ~23.0km altitude and a large umbrella spreading at 18.0–18.5km, according to the side view (Fig. 8b). Thus, the plume rises at a fairly rapid average speed of ~20 m s$^{-1}$. For this thick and opaque plume, the dark pixel $BT_{11}$ of 197.3K leads to a radiometric height range of 15.8–18.3km, the upper end of which agrees well with the geometric umbrella height estimate.

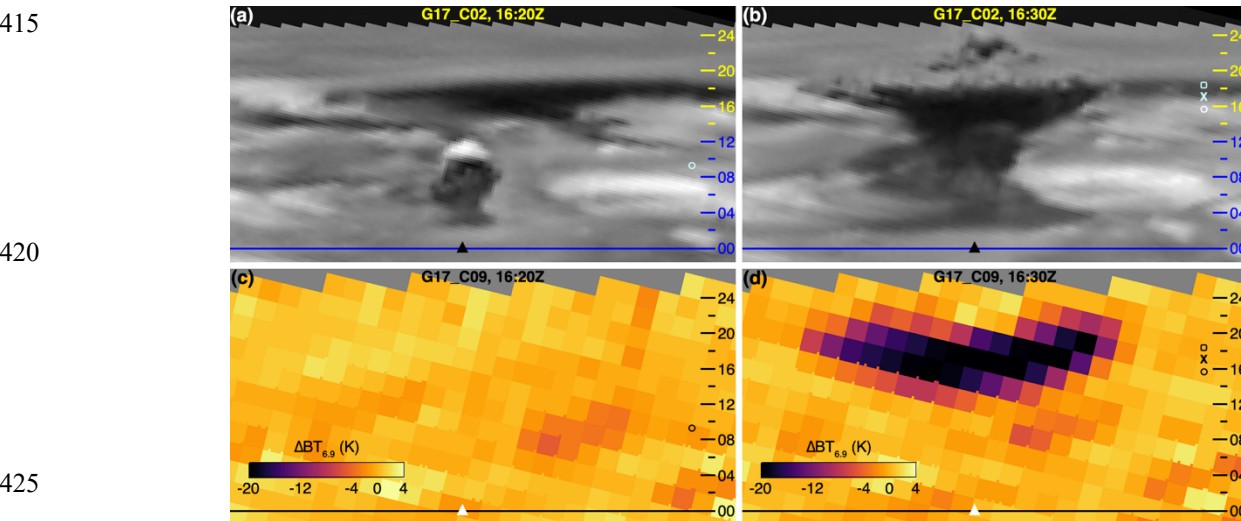

**Figure 8.** The eruption plume on 10 April 2021 at **(left)** 16:20UTC and **(right)** 16:30UTC in GOES-17 imagery: **(top)** channel 2 and **(bottom)** channel 9 running difference, with elevation markings as in Fig. 5.

Here we note that this eruption almost reached the 24km maximum height measurable with the side view technique at La Soufrière's location. Above that height, the plume would have been cut off by the limb mask currently applied to ABI images by NOAA. We recommend retaining space pixels in future ABI data releases to avoid such limitations and also to ensure consistency with Himawari-8 imagery, which smoothly transitions into space.

The plume was generally difficult to identify in any of the IR channels; in fact, at 16:20UTC the column rising in the low/mid troposphere could not be identified at all (Fig. 8c). At 16:30UTC, the upper part of the umbrella above ~12km did appear as an area of slightly reduced temperatures; however, the contrast was low against a cold background caused by a fairly moist atmosphere and the significant presence of clouds and suspended ash, especially towards the limb. We found that the umbrella could be best discerned in the channel 9 (6.9μm mid-level water vapor band) running difference, obtained by differencing the 16:30UTC and 16:20UTC images (Fig. 8d). Here, the pattern of negative temperature differences has a centerline at ~18km, consistent with the umbrella height deduced from the visible image.

This case exemplifies that the IR channel optimal for plume identification varies with the atmospheric temperature and moisture profile, and that change detection can be aided by the computation of running differences when multitemporal imagery is available. We further explore this issue in the next section.

### 3.3 11 April, 10:45–11:00UTC

This explosion produced a mushroom cloud, which reached 18.0–18.5km altitude according to the side views (Fig. 9a and Fig. 9b, Supplement Animation 3). The corresponding dark pixel $BT_{11}$ (~200.0K) imply radiometric heights of 14.7–19.4km. The geometric height falls between the midpoint and upper end of this height range. The umbrella can be identified at ~18km

altitude in the IR side views too (Fig. 9c and Fig. 9d). In this case, however, lower parts of the eruption column down to 7–8km could also be observed, reflecting background conditions (moisture, clouds, ash) different than encountered in the previous examples.

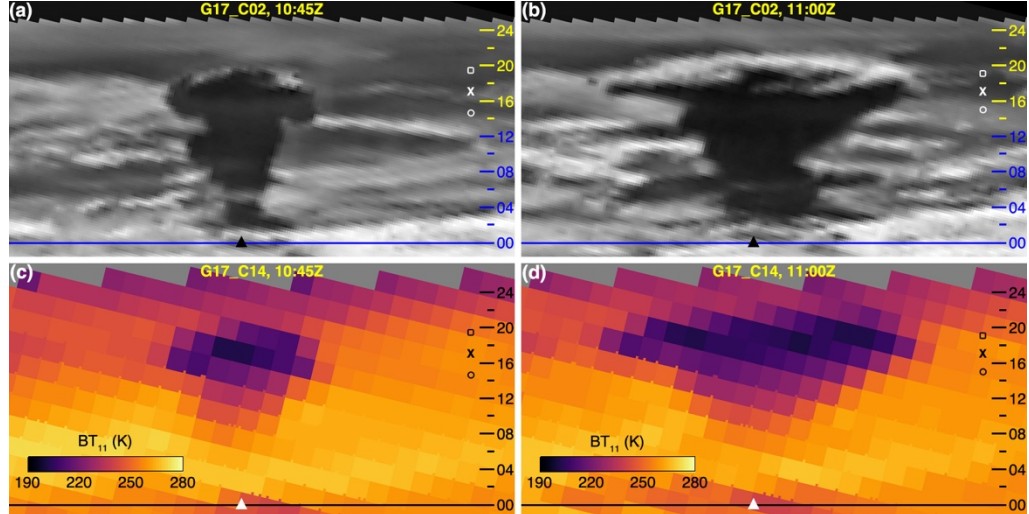

**Figure 9.** The eruption plume on 11 April 2021 at **(left)** 10:45UTC and **(right)** 11:00UTC in GOES-17 imagery: **(top)** channel 2 and **(bottom)** channel 14, with elevation markings as in Fig. 5.

This prompted us to compare the side views of the 10:45UTC plume in all nine ABI IR channels. In Fig. 10, the color scale is stretched individually for each channel between the minimum and maximum brightness temperatures of the scene. In the water vapor bands (channels 8, 9, and 10), only the top of the plume is recognizable. As the altitude of the water vapor weighting function's peak decreases from band 8 to band 10, slightly more of the umbrella becomes discernible, but detection generally is limited to heights above ~12km. In the rest of the IR channels, which are less affected by water vapor absorption, lower parts of the plume down to 7–8km are also observable, with slight differences in detectability between bands. Bands 12 and 16, however, show noticeably increased noise as a consequence of the loop heat pipe anomaly.

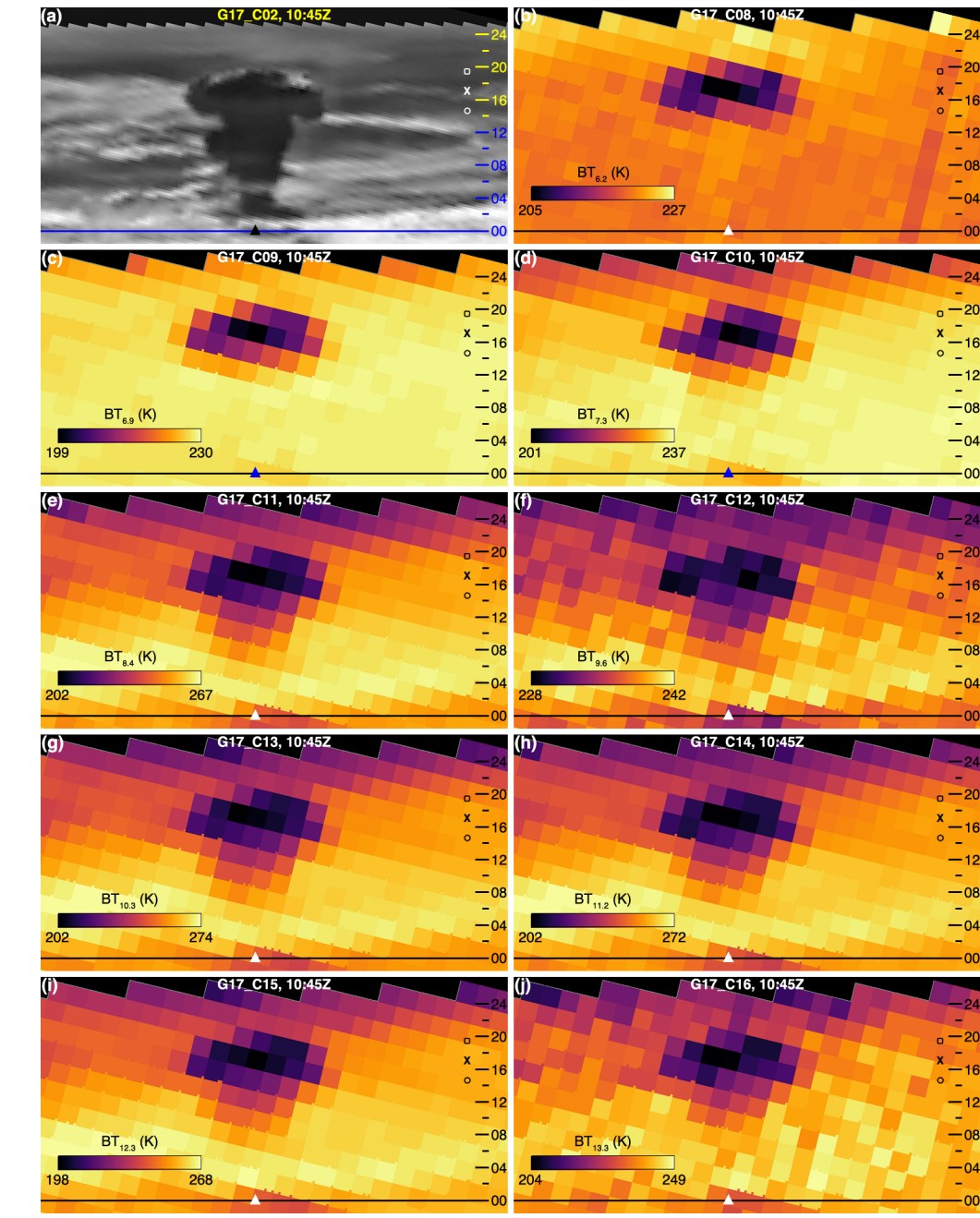

**Figure 10.** The eruption plume on 11 April 2021 at 10:45UTC in GOES-17 imagery: **(a)** the visible channel 2 and **(b to j)** the IR channels 8–16.

### 3.4 11 April, 13:30–13:40UTC

This was the largest of the daytime eruptions. In the 10-minute slot ending at 13:30UTC, the column reached an altitude of 16.0–16.5km in the GOES-17 image, rising with an average speed of ~27 m s$^{-1}$ (Fig. 11a, Supplement Animation 4). The likely warm-biased dark pixel $BT_{11}$ of 216.2K corresponds to a wide radiometric height range of 12.8–23.5km, the lower bound of which underestimates the geometric height by more than 3km. Here the mean of the radiometric height solutions (18.1km) is a better match to the near-tropopause geometric height. By 13:40UTC the plume formed an umbrella at 18.5–19.0km (Fig. 11b).

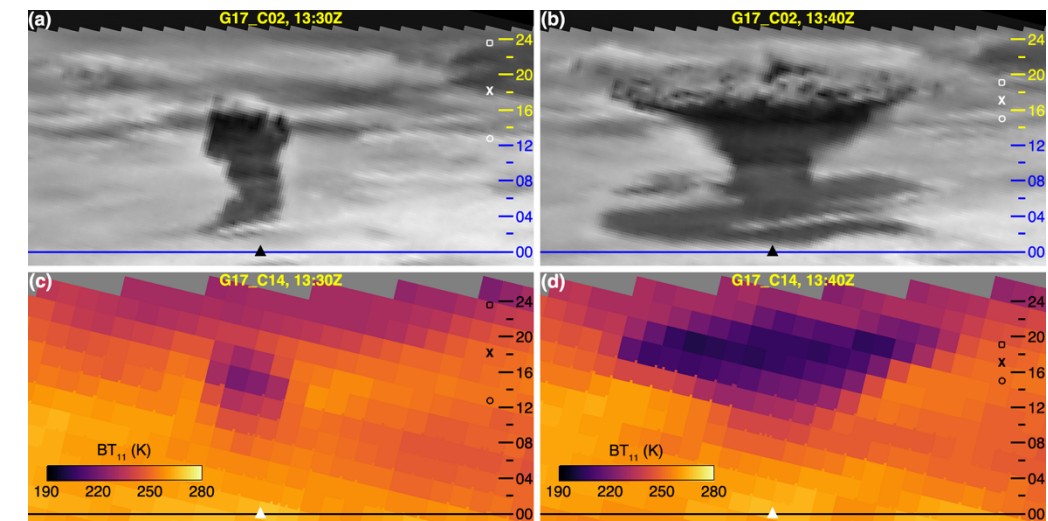

**Figure 11.** The eruption plume on 11 April 2021 at **(left)** 13:30UTC and **(right)** 13:40UTC in GOES-17 imagery: **(top)** channel 2 and **(bottom)** channel 14, with elevation markings as in Fig. 5.

For this thick opaque plume top, the upper bound of the 15.0–19.1km radiometric height range, obtained from a dark pixel temperature of 199.6K, is in excellent agreement with the geometric height estimate. Note that the plume tops can be identified in the IR side views at approximately the same altitude as in the visible side views (Fig. 11c and Fig. 11d).

### 3.5 13 April, 10:30–10:45UTC

An extensive layer of cirrus (Ci) clouds covered the area during this eruption. At 10:30UTC, the dark contours of the rising column can be faintly seen through the veil of Ci, which is accentuated by the long air path of the side view (Fig. 12a, Supplement Animation 5). The plume top location is difficult to determine precisely, but it is still below the Ci at approximately 10–11km altitude. The single radiometric height solution of 5.4km, corresponding to a dark pixel $BT_{11}$ of 269.2K, is a significant underestimate.

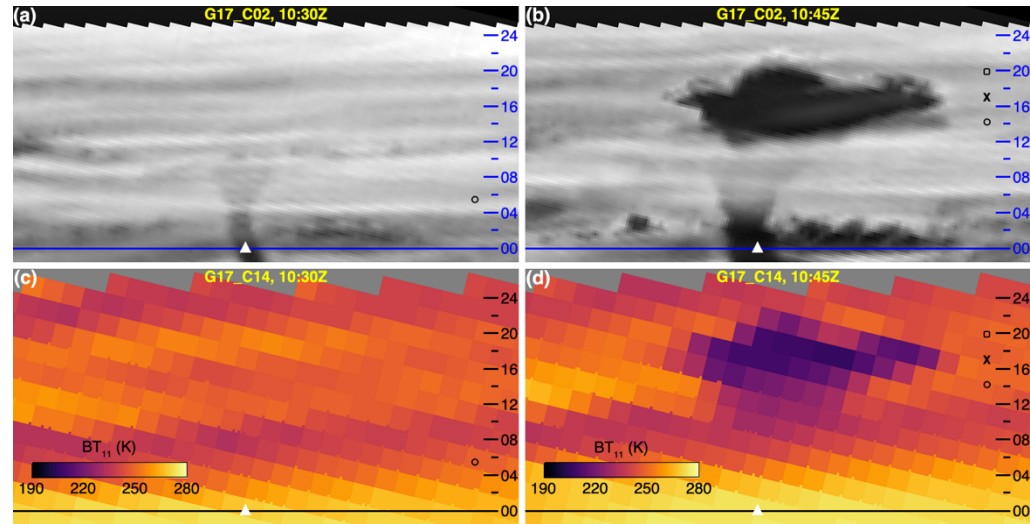

**Figure 12.** The eruption plume on 13 April 2021 at **(left)** 10:30UTC and **(right)** 10:45UTC in GOES-17 imagery: **(top)** channel 2 and **(bottom)** channel 14, with elevation markings as in Fig. 5.

By 10:45UTC, the plume breached the Ci layer and featured an umbrella spreading at 17.0–17.5km with an OT at ~20km (Fig. 12b). Note that the Ci intersects the plume at an altitude of 12–13km, which agrees well with the Ci heights retrieved from GOES–MODIS stereo pairs (see Sect. 3.7). The minimum $BT_{11}$ of 203.4K implies a radiometric height range of 14.2–19.8km, the midpoint of which is a good match to the geometric umbrella height.

The thicker strands of Ci appear as horizontal stripes of colder temperature in the IR side views (Fig. 12c and Fig. 12d). The growing column is undetectable in band 14 (or in any other IR band) at 10:30UTC. In the 10:45UTC IR image, however, the above-Ci umbrella and OT can be both located at about the same height as in the visible side views.

### 3.6 22 April, 15:10–15:20UTC

Our final example was the last eruption in the current series, which produced a relatively small and fully tropospheric plume. The atmosphere was noticeably drier and clearer on this day, with less haze, only low-level clouds, and no suspended ash from prior eruptions (the penultimate small explosion occurred four days earlier on 18 April). The height of the eruption

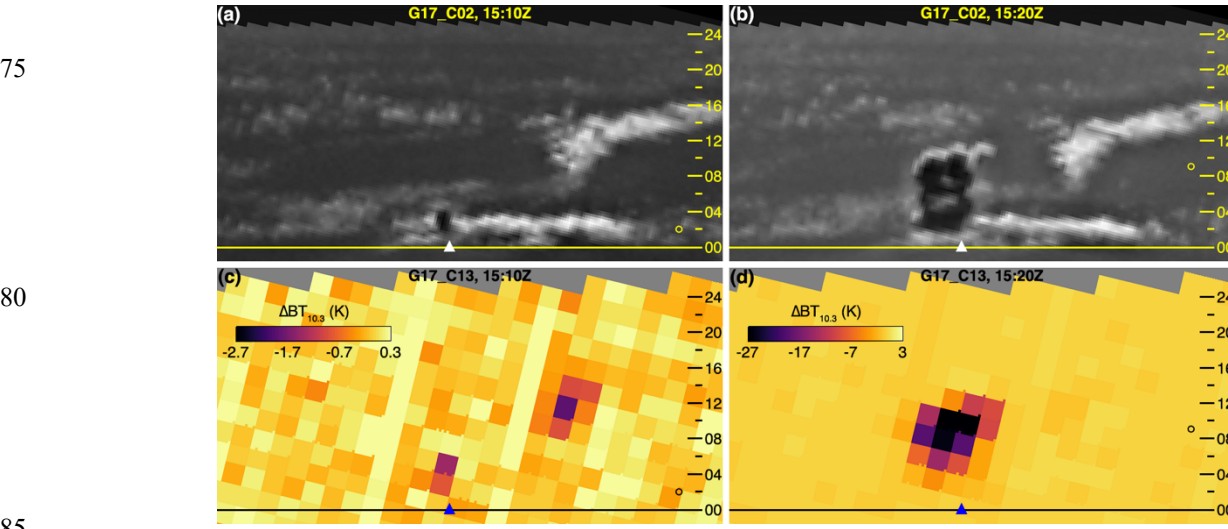

**Figure 13.** The eruption plume on 22 April 2021 at **(left)** 15:10UTC and **(right)** 15:20UTC in GOES-17 imagery: **(top)** channel 2 and **(bottom)** channel 13 running difference, with elevation markings as in Fig. 5.

column increased from 4.0km to 10.5–11.0km between 15:10UTC and 15:20UTC, as determined from the visible side views (Fig. 13a and Fig. 13b, Supplement Animation 6). In both time slots there was a single radiometric height solution, increasing from 2.0km ($BT_{11,min}$ = 286.9K) to 9.0km ($BT_{11,min}$ = 244.1K) and thus having a low bias of ~2km.

This case also demonstrated that under sufficiently clear and dry conditions, even small plumes can be detected (and at the correct height) in the IR side views. As shown in Fig. 13c and Fig. 13d, practically the entire eruption column all the way down to the vent could be identified in the channel 13 (10.3μm) running difference images. The "clean" IR longwave window band worked particularly well here, because it is the least sensitive among the IR window bands to water vapor.

## 3.7 GOES-16–MODIS stereo retrievals and CALIPSO lidar profiles

The MODIS Terra and MODIS Aqua instruments imaged La Soufrière on 10 April at 14:36UTC and 17:42UTC, respectively. By that time the ash from prior eruptions had spread hundreds of kilometres east and also expanded in the north-south direction, forming a triangle-shaped volcanic cloud. There were 23 eruptions before the Terra overpass and two eruptions between the Aqua and Terra overpasses, including the large explosion discussed in Sect. 3.2.

As shown in Fig. 14a and Fig. 14c, the brownish ash layer was observed against the background of white meteorological clouds. The crescent-shaped Ci bands likely indicate modulation by gravity waves emanating from the explosions. The interpretation of retrievals in such a complex multi-layer scene requires caution. The 3D Winds algorithm (Carr et al., 2019) tracks targets (6×6 km$^2$ image chips in this case) without classifying their type; therefore, the height and motion retrievals plotted in Fig. 14b and Fig. 14d contain both ash and cloud targets.

Comparisons with lidar measurements revealed that stereo matchers generally track the lower layer in a two-layered scene when the top layer's optical depth <~0.3 (see Mitra et al., 2021 for a recent study). In the semi-transparent parts of the

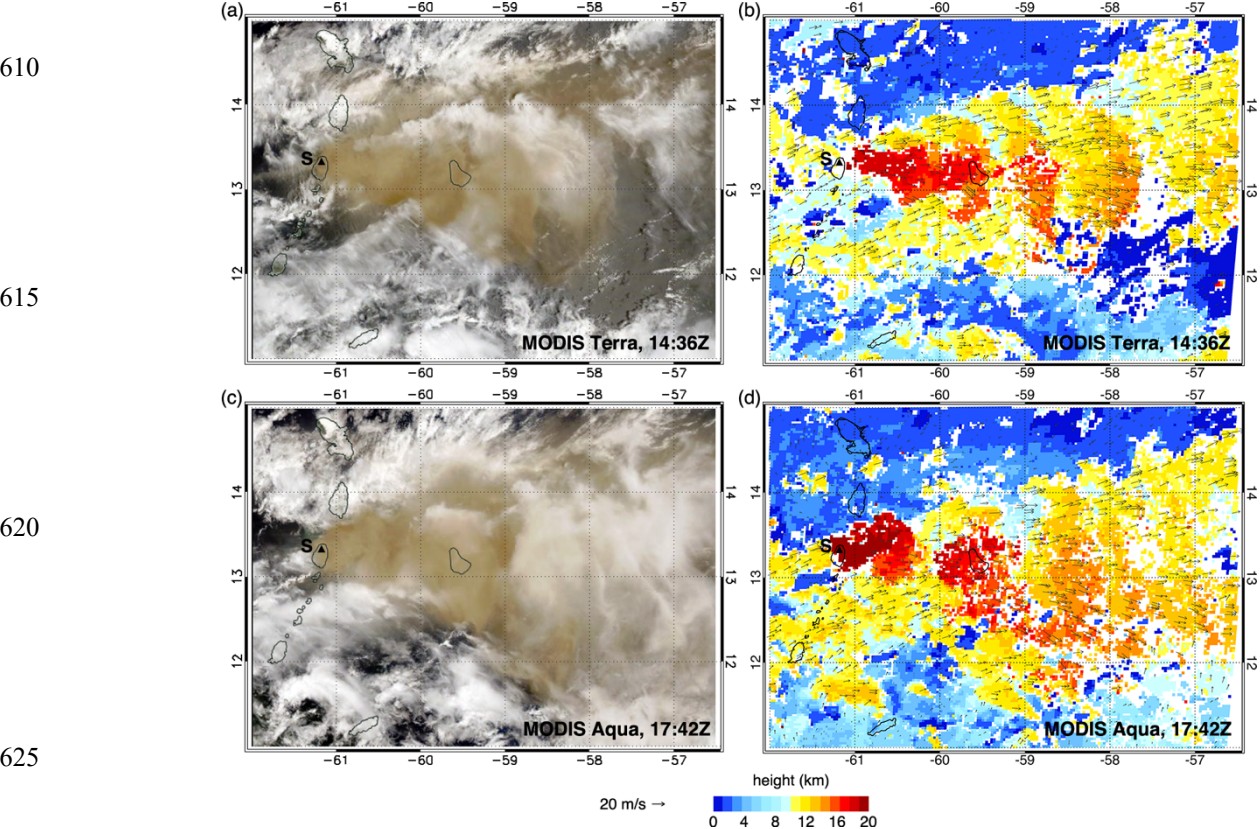

**Figure 14.** True color image of the plume on 10 April 2021 by **(a)** MODIS Terra (14:36UTC) and **(c)** MODIS Aqua (17:42UTC). The corresponding GOES-16–MODIS 3D Winds stereo heights are plotted in panels **(b)** and **(d)**, with motion vectors shown for a random 5% of retrievals. La Soufrière is marked by the black triangle and letter 'S'.

ash layer, the algorithm tracks the lower-level meteorological clouds, which have more texture and contrast. The stereo retrievals in ash-free areas indicate Ci up to 12–13km altitude (yellow hue), which agrees well with the side view Ci height estimate in Sect. 3.5. From this we conclude that 3D Winds heights above 13km (orange or more reddish hue) can confidently be classified as ash.

The maximum stereo-retrieved plume height is 22.9km for both the Terra and Aqua scenes, which is in good agreement with the largest OT heights obtained from the GOES-17 side views. Both scenes show a general decrease in height as the ash was advected east by westerly winds of 15–20 m s$^{-1}$. The plume height immediately east of the volcano was 17–18km during the Terra overpass. During the Aqua overpass, however, the plume east-northeast of the volcano was at a higher altitude of

19–21km, which was undoubtedly the result of the powerful explosion that occurred at 16:30UTC (see Sect. 3.2). By the time the plume reached Barbados, its height subsided to 16–17km. Near longitude 58ºW the retrieved plume height reduced to 14–15km and even further east the stereo retrievals started to pick up the height of the Ci as the plume became too tenuous to track, although the true color images still indicate the presence of a thin ash layer that reduces the brightness of the white clouds underneath. Overall, these stereo plume heights are in good agreement with the near-field plume heights derived previously from the side views.

The CALIPSO satellite unfortunately did not fly over the volcano. However, there were 13 CALIPSO orbits between 10–13 April that intersected the far-field plume as it drifted east-northeast across the Atlantic Ocean. Cloud-Aerosol Lidar with Orthogonal Polarization (CALIOP) backscatter profiles indicated volcanic particles between 5–20km, which is generally consistent with the height range of the side view retrievals (see Table S1 in the Supplement). A more detailed analysis of the CALIOP profiles obtained closest to La Soufrière is given in Appendix A.

## 4 Discussion

### 4.1 Overview of all daytime height retrievals

The height retrievals for all 30 analyzed daytime cases are plotted in Fig. 15 with the actual data listed in Table S1 in the

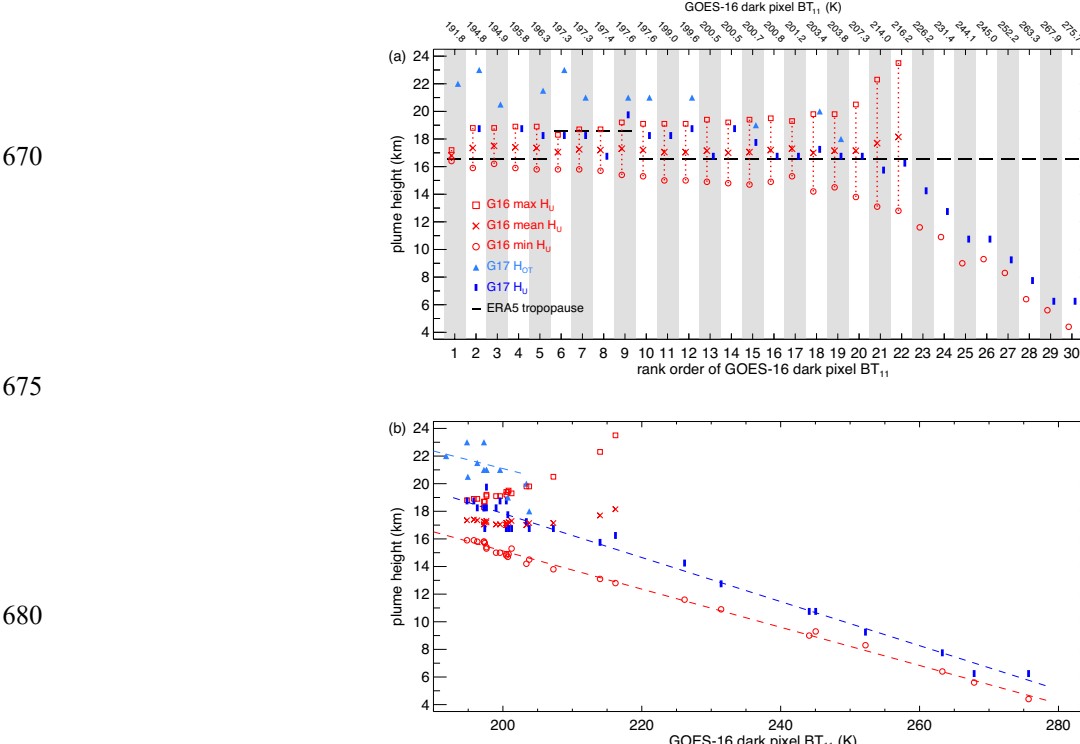

**Figure 15.** Comparison of GOES-17 side view heights (blue) and GOES-16 temperature-based heights (red) as a function of GOES-16 dark pixel $BT_{11}$: **(a)** rank order and **(b)** absolute value. $H_U$ and $H_{OT}$ are the height of the umbrella and the overshooting top, respectively; $H_{OT}$ is only estimated from GOES-17. The height of the ERA5 cold point tropopause is also indicated. In panel **(b)**, the dashed lines are linear fits to the GOES-17 umbrella and OT heights (cases 15, 18, 19 are excluded for OT).

Supplement. For the easy comparison of the geometric and radiometric heights in individual cases, the results are indexed and plotted in Fig. 15a according to the rank order of GOES-16 dark pixel $BT_{11}$. For plume temperatures warmer than 220K, the single radiometric height always underestimates the side view height. These cases represent smaller fully tropospheric eruptions (e.g. Sect. 3.6) or the growing phase of larger eruptions that later reach the stratosphere (e.g. Sect. 3.2). For colder plume temperatures between 200–220K, the mean of the upper and lower radiometric height solutions tends to agree best with the geometric height. In these cases, the umbrella spreads near the tropopause. For the coldest plumes below 200K, which are also the tallest, the stratospheric radiometric height solution is usually a fairly good match to the geometric height. The dark pixel $BT_{11}$, however, is not a particularly good predictor of the maximum OT height. In the three instances when the side view OT height is within the radiometric height range (index 15, 18, 19), either a still growing or an already collapsing OT was observed. This sampling bias is the consequence of the OT reaching its maximum altitude in between 10-minute FD scans.

The same height retrievals are plotted in Fig. 15b versus the absolute value of dark pixel $BT_{11}$. The -6.3K/km tropospheric lapse rate derived from the GOES-17 geometric heights and GOES-16 brightness temperatures is comparable to the ERA5 lapse rate of -7.2K/km. However, the observed $BT_{11}$ shows a warm bias of 10–20K due to semitransparency and/or subpixel effects. The cluster of points characterized by geometric heights of ~17km and a range of brightness temperatures between 197–207K likely represents varying degrees of semitransparency-related warm bias in thinning umbrellas spreading near the tropopause.

The overshooting tops are in apparent thermal disequilibrium, being 10–20K colder than the stratospheric ambient, when they are assumed to be characterized by the minimum $BT_{11}$, as is usually done. In fact, the OTs seem to cool with an effective above-tropopause lapse rate of -7.8K/km, which is essentially the upper tropospheric ERA5 lapse rate; however, sample number is small and the height–temperature correlation is poor (-0.3). It might be better to characterize OTs by the maximum $BT_{11}$, provided a well-defined local maximum such as a central warm spot within a cold ring can be identified in the plume. This is not always the case and the OT location might not even coincide with either the minimum or the maximum plume temperature. Additional complicating factors include decompression cooling and brightness temperature biases due to semitransparency and/or subpixel effects. The non-trivial problem of linking OTs to the complex and rapidly changing temperature structure of volcanic plumes is deferred to a later study, which can take advantage of the 1-minute sampling offered by the MESO2 scans.

In a final summary, Fig. 16 plots the GOES-17 side view height against the best-match temperature-based height. For relatively warm tropospheric eruption columns, the single radiometric height underestimates the geometric height by 2–3km with an overall low bias of -1.6km. For umbrellas spreading near the tropopause, the mean of the radiometric height solutions is a reasonable approximation to the geometric height typically within ±1km and with an overall high bias of +0.6km. For the coldest and tallest umbrellas, the temperature-based stratospheric height agrees well with the geometric height, showing deviations within ±0.8km and an overall high bias of +0.3km. Such a good agreement suggests only small biases (thermal disequilibrium, semitransparency/subpixel effects) in the brightness temperature measured in optically thick, opaque and non-violently spreading plumes. However, the stratospheric height solution corresponding to the dark pixel temperature always underestimates the maximum OT height by up to 5km, with an overall low bias of -2.9km in our dataset.

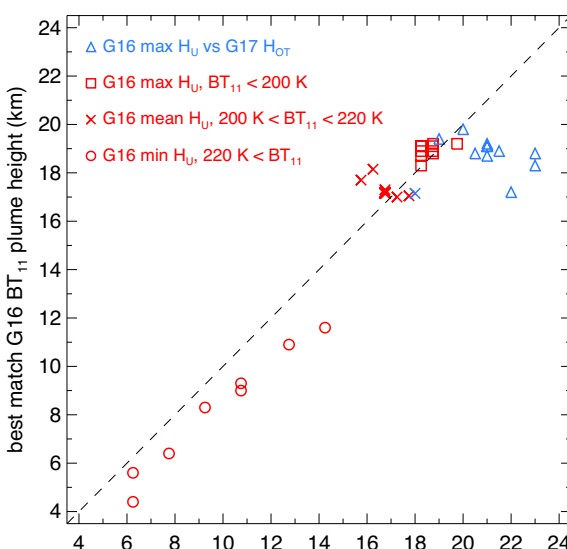

**Figure 16.** GOES-17 side view height versus the best match GOES-16 temperature-based height. The GOES-17 OT height is plotted against the GOES-16 maximum (stratospheric) height solution. The dashed line is the 1:1 line.

Because the atmospheric temperature structure showed little diurnal or day-to-day variations during the entire eruptive period (see Fig. 4), the results from the above comparison of daytime height retrievals might also be useful to 'calibrate' the temperature-based height estimates for the numerous night-time eruptions. The night-time dark pixel brightness temperatures can be classified into one of three categories found for the daytime cases ($BT_{11} > 220K$, $200K < BT_{11} < 220K$, or $BT_{11} < 200K$) to select the corresponding bias-corrected best-match (min, mean, or max) radiometric height solutions.

**4.2 Comparison with the April 1979 eruptions**

Noting the similarities in measurement techniques, atmospheric conditions, and eruption heights, we briefly review La Soufrière's last major eruptions, which occurred between 13–25 April 1979, the most intense one on 17 April. The atmospheric temperature profile resembled the current case, with a cold point tropopause of 193K between 16.2–17.0km (Barr and Heffter, 1982). The plumes were fairly well observed by both aircraft and satellite. In a direct analogue to our
method, the height of the large plume on 17 April was determined from a side view photograph taken by an aircraft six minutes after the explosion from a distance of 104km. The estimates yielded a plume top at 18–20km altitude (Fiske and Sigurdsson, 1982). Airborne lidar measurements collected between 17–19 April detected distinct stratospheric ash layers at 16, 17, 18, and 19.5km (Fuller et al., 1982). Height estimates for 17 April were also obtained from 11-μm brightness temperatures from the SMS-1 (Synchronous Meteorological Satellite-1) geostationary satellite located at 70ºW, which
indicated a stratospheric plume at 18km altitude (Krueger, 1982). Maximum plume heights generally varied between 10–

20km during the entire eruption period. Overall, the observed plume heights of the 1979 and the current series of eruptions were very similar, suggesting a comparable level of activity.

**5 Summary**

We presented daytime plume height estimates for the April 2021 La Soufrière eruptions obtained from GOES-17 side views
and GOES-16–MODIS stereo views. Our side view estimates indicated that only a couple of eruptions remained fully in the troposphere, typically between 6–14km. Most of the plumes, however, either spread at the tropopause near 16–17km or penetrated the lower stratosphere reaching altitudes between 18–20km. Overshooting tops up to 23km altitude were also observed in the largest explosions. The independent stereo retrievals for the Terra and Aqua overpasses on 10 April also showed maximum plume heights of 23km and a main spreading layer of 18–21km, confirming the side view results. By the
time the visible ash cloud reached Barbados, its altitude decreased to 16–17km. We note that the plume heights measured during the current eruptions were very similar to the ones observed during the volcano's last major eruptions in April 1979.

The geometric heights were compared to the radiometric height or height range corresponding to the measured dark pixel plume temperature (minimum $BT_{11}$). For smaller eruption columns, the single radiometric height underestimated the geometric height by a couple of kilometers due to a warm bias of 10–20K, caused mostly by subpixel effects. For plumes
spreading near the tropopause, the midpoint of the radiometric height range was a reasonable approximation to the geometric height. This was so because for the tropical temperature profile of La Soufrière, the average of the upper and lower radiometric height solutions is near the tropopause, due the tropospheric and stratospheric lapse rates being of opposite sign but comparable magnitude. The methods were most consistent in the coldest umbrellas, where the upper bound of the radiometric height range (stratospheric solution) agreed well with the geometric height, indicating small brightness
temperature biases in the optically thickest plumes. These three plume classes were fairly well separated by brightness temperature thresholds; thus, the daytime height comparison results could be used to 'calibrate' and bias correct the night-time radiometric height retrievals.

Although the side view method was originally developed for the highest resolution visible red band images, we have shown in the current work that depending on channel and atmospheric conditions, plume heights can also be estimated from
IR side views, albeit with larger uncertainty (±2km per ±1pixel). Due to increased water vapor absorption along the long view path, plume detection in IR side views typically works only above ~12km; however, in dry and clear atmospheres, smaller plumes can occasionally be identified too. These results suggest that the side view technique can provide useful complementary height retrievals during night time, especially for larger plumes.

On a final note, we believe that obtaining higher frequency side view imagery of a volcanic eruption near the limb of the
GOES-R Earth scan would be beneficial in the future. The full disk oblique imagery used in the current study only offers 10-minute sampling; however, positioning an ABI MESO domain over a near-limb volcano would provide 1-minute side view imaging. The improved temporal sampling of a rapidly rising eruption column would allow to better capture the maximum

height attained by the plume and would also provide unique data for the study of volcanic jet dynamics, comparable to the side view imagery obtained in laboratory water tank experiments on particle-laden jets (Gilchrist and Jellinek, 2021).

**Appendix A: Comparison with CALIOP profiles on 10 April, 06:25–06:26UTC**

Here we analyze the track that passed closest to the volcano (~100km east of La Soufrière) on 10 April between 06:25–06:26UTC (level 1 data file CAL_LID_L1-Standard-V4-11.2021-04-10T06-10-21ZN.hdf). This night-time (descending) track is overlaid on the 06:30UTC GOES-16 $BT_{11}$ image in Fig. A1a and the corresponding 532nm total attenuated backscatter profiles are plotted in Fig. A1b. As shown, there was a stratospheric layer stretching between 12.0ºN–14.1ºN and reaching heights up to ~18.5km.

These lidar layer heights are consistent with preceding night-time eruption heights derived from the dark pixel $BT_{11}$ and calibrated by the daytime geometric–radiometric height comparison discussed in Section 4.1. There were nine night-time eruptions on 10 April before 06:25UTC. The last one prior to the CALIPSO overpass occurred between 05:20–05:40UTC and had dark pixel plume temperatures of 194.2–197.0K. Using the 'calibration' in Fig. 16, these temperatures correspond to best-match (stratospheric) radiometric heights of 18.1–18.9km. The coldest plume temperature in the 06:30UTC GOES-16 image was 193.8K, resulting in a 'calibrated' radiometric height of 17.9km; again, in good agreement with the lidar heights.

The general comparison between radiometric and lidar heights shows strong similarities with the comparison between radiometric and side view heights. Fig. A1b reveals a complex vertical structure of multiple cloud and ash layers, the higher of which are likely often semitransparent. A $BT_{11} > 220$K yields a single radiometric height that is either in between layers or represents a lower optically dominant layer. For $220$K $> BT_{11} > 200$K (13.12ºN and 13.42ºN), the mean of the stratospheric and tropospheric radiometric height solutions is in reasonable agreement with the near-tropopause lidar height.

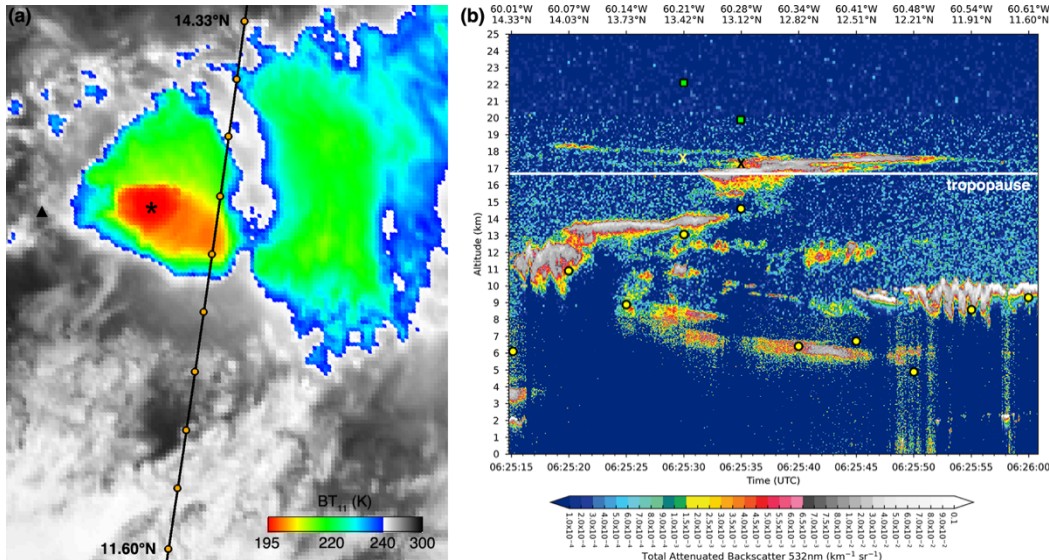

**Figure A1. (a)** GOES-16 $BT_{11}$ of the plume on 10 April 2021 at 06:30UTC. The black line is the CALIPSO orbit track with 10 specific points marked by orange dots. The black triangle is the volcano and the black star is the coldest plume pixel. **(b)** CALIOP total attenuated backscatter at 532nm. The white line is the tropopause, and the circles, crosses, and squares respectively depict the minimum, mean, and maximum plume heights corresponding to the $BT_{11}$ at the 10 orange track points in panel **(a)**.

*Data availability.* The GOES-R ABI L1B radiances are available from the NOAA Comprehensive Large Array-data Stewardship System (CLASS) archive (https://doi.org/10.7289/V5BV7DSR, GOES-R Calibration Working Group and GOES-R Series Program, 2017). There are no restrictions on the use of GOES-R data (https://registry.opendata.aws/noaa-goes/). The CALIPSO Lidar Level 1 Version 4.11 Data Product is available free of charge from the NASA Langley Research Center Atmospheric Science Data Center (https://asdc.larc.nasa.gov/data/CALIPSO/LID_L1-Standard-V4-11/). The lidar profiles were plotted with the open source command-line program ccplot, available at https://ccplot.org. The open source Fiji image processing package is available at https://imagej.net/software/fiji (last access: 6 April 2022).

*Data and video supplement.* The 3D Winds stereo retrievals and all mentioned animations are available in the Supplement.

*Supplement.* The supplement related to this article is available online at:

*Author contributions.* ÁH developed the idea and methodology of the side view retrievals during discussions with GAH and SAB. Retrievals from the 3D Winds stereo code were provided by its developers JLC and DLW. ÁH and JB analyzed the results. ÁH prepared the manuscript with significant contributions from all authors.

*Competing interests*. The authors declare that they have no conflict of interest.

*Acknowledgements*. ÁH, JB, GAH, and SAB are members of the VolPlume project within the research unit VolImpact funded by the German Research Foundation DFG (FOR 2820). This work also contributes to the Cluster of Excellence "CLICCS—Climate, Climatic Change, and Society" funded by the Deutsche Forschungsgemeinschaft DFG (EXC 2037, Project Number 390683824), and to the Center for Earth System Research and Sustainability (CEN) of Universität Hamburg.

*Financial support*. This research has been supported by the Deutsche Forschungsgemeinschaft (grant nos. FOR 2820 and EXC 2037, project number 390683824).

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
