# Peer review of "Measurement report: Plume heights of the April 2021 La Soufrière eruptions from GOES-17 side views and GOES-16–MODIS stereo views"

_Atmospheric Chemistry and Physics, 2022_

## Author Comment (AC1)

We thank the Referees for their constructive comments. In the following, we give a point-by-point response to each of the issues raised.

**Referee #1**

*Review of "Measurement report: Plume heights of the April 2021 La Soufriere eruptions from GOES-17 side views and GOES-16-MODIS stereo views" by A. Horvath et al.*

*General Comments*

*In this new study, Horvath et al. present a comparison of plume height estimates of a series of eruptions of La Soufriere in April 2021. The study compares the results of different measurement techniques and methods to estimate the plume heights, including side views and stereoscopic satellite imaging in both, visible and Infrared spectral bands as well as plume height estimation based on the minimum brightness temperature of the volcanic clouds. The study provides practical advice on how to interpret and possibly resolve ambiguities in height estimation related to the brightness temperature method.*

*Overall, this is an interesting and carefully conducted study, which fits well into the scope of ACP. The manuscript is very clear and concise. I would like to recommend the paper for publication in ACP subject to just a few minor corrections and clarifications as listed below.*

*Specific Comments*

*l41-43: Although it is a classical method, can you provide some references/examples of earlier studies using the brightness temperature method for volcanic plume height estimation?*

We added three references: de Michele et al. (2019), Oppenheimer (1998), and Prata and Grant (2001). We included the first of these per the request of Referee #2, because, like us, de Michele et al. (2019) also uses the term "dark pixel temperature" as a synonym for "the minimum 11μm brightness temperature".

*l90-91: It would be good to add a reference for the ERA5 data:*

*Hersbach, H, Bell, B, Berrisford, P, et al. The ERA5 global reanalysis. Q J R Meteorol Soc. 2020; 146: 1999–2049. https://doi.org/10.1002/qj.3803*

Done.

*l603-605: Based on recent findings from the Tonga eruption, I was also wondering why one would always necessarily pick the minimum brightness temperature of a volcanic plume to estimate the plume height. For the Tonga eruption, this approach would estimate a layer height close to the tropopause and significantly underestimate the heights of the upper layers of the plume. Upper layers would have to be estimated by inspecting the maximum brightness temperature near the center of the cloud, as suggested here. Perhaps, this aspect could be mentioned/discussed more early in the paper, when first introducing the brightness temperature method.*

We included a discussion on this in the last paragraph of Section 2.2, as suggested.

*l641-651: This paragraph provides some interesting information on the 1979 eruption of La Soufriere, but it became not so clear to me, how it relates to the present study? Do you like to compare earlier methods for plume height estimate with present state-of-the-art methods in general?*

This section simply notes the similarities between the eruptions in terms of atmospheric conditions, plume heights, and measurement techniques. For example, the side view aerial photo measurements are worth acknowledging. The similar plume heights also suggest a comparable level of activity. We made some edits to hopefully better convey the purpose of this paragraph.

*Supplements: The videos/animations for the different eruptions are very nice, but could you possibly make them run a bit more slowly? (On my notebook, each of the animations takes only 2-3 s to run.)*

We increased the time delay between the frames by a factor of 3–4, thus, the duration of the animations is now 6–8 seconds.

*Technical Corrections*

*l271: fix/explain "put oft-present cirrus"?*

Changed to "indicated cirrus".

*l353: "a nice mushroom cloud" -> "a mushroom cloud"*

Done.

---

## Author Comment (AC2)

We thank the Referees for their constructive comments. In the following, we give a point-by-point response to each of the issues raised.

**Referee #2**

*Review of Horvath et al. "Measurement report: Plume heights of the April 2021 La Soufrière eruptions from GOES-17 side views and GOES-16–MODIS stereo views"*

*Reviewer: Mike Fromm*

*Horvath et al. apply a suite of techniques they introduced in a pair of 2021 publications to the stratospheric volcanic eruption at Soufriere St. Vincent in April 2021. This volcano and eruption event were advantageous for its location (quite close to perfect limb view) and the dozens of eruption pulses spanning a variety of column heights. In this manuscript the authors expand on their previously introduced topic of side-view IR limb profiling of the eruption column, demonstrating promise for eruption-monitoring applications at night.*

*Like their pair of publications in 2021, this manuscript achieves a high quality in terms of rigor, clarity, and organization. Figures are clear and easy to interpret. For the most part, referenced literature is appropriate. Sources of uncertainty are fairly described and dealt with. I have only minor questions and suggestions for improvement. Given an adequate response to these, I expect to highly recommend publication.*

*The overriding weakness of this manuscript, minor though it is, is that the reader may want to know if the innovative analyses herein have a relevance beyond the arcane or academic. Presumably the answer is yes, but the authors do not "sell" their advances in terms of practical applications. For instance, it would be great to learn where on Earth these side views intersect with volcanoes. Each operational Geo bird has an entire limb perimeter to offer. Hence, some accounting for each Geo bird's limb swath and the volcanoes worth watching within each swath would immeasurably add to the impact of this paper.*

We added a figure (the new Fig. 3) with the corresponding discussion to Section 2.1, which plots the limb swaths of the currently operational geostationary satellites and the volcanoes within these regions. Following Horváth et al. (2021a), limb regions are defined as VZA>80º; however, we also plot volcanoes observed under a relaxed constraint of 80º>VZA>60º. As shown, most of the major volcanic regions are observed at oblique angle by at least one of the GEO satellites.

*Secondarily, it is serendipitous that La Soufriere was so close to a Geo/ABI limb. The paper makes a wonderful demonstration of that near ideal condition. However, from a volcano reference frame, it would be important for the reader to get some idea as to how these side-view visible and IR potentials degrade away from the limb toward a satellite's nadir. The authors are encouraged to present an assessment, within a volcano-location-with-respect-to-Geo-limb reference frame, how far from limb these tools can apply. I realize this is a multi-variate challenge, but it is nonetheless important for the reader to know how much value is delivered by this proposed Measurement Report.*

We added a figure (the new Fig. 2), which sketches the side view measurement principle and demonstrates the increase of foreshortening at smaller VZA. The discussion in Section 2.1 was also expanded accordingly. The height error caused by a 1-pixel error in the location of the plume point above the vent increases as VZA decreases, which is manifested by more densely packed true height isolines. The minimum VZA at which the technique is still useful is a judgment call. For example, at a VZA of 22º, a 1 visible pixel location error causes a 1.3km height error. Identifying the above-vent plume point is also more difficult at small VZA, because the view is more overhead and thus captures the spreading umbrella rather than the side of the column. It's difficult to put a hard threshold on VZA, but in our experience height retrievals are still feasible for eruptions observed at VZA>60º (e.g., Sangay). Hence the choice of the relaxed constraint of 80º>VZA>60º when plotting limb volcanoes in the new Fig. 3 discussed above.

*I was curious to know if there were any strategic encounters by the lidar aboard CALIPSO? If so, might the authors consider showing such an encounter and discussing agreement/disagreement with their independent plume height calculations? If there were no such encounters, there is no reason to modify the paper.*

While CALIPSO did not fly over the volcano, there were 13 orbits between 10–13 April that intersected the far-field plume as it drifted east across the Atlantic Ocean. CALIOP backscatter profiles indicated volcanic particles between 5–20km, in general agreement with the height range of side view retrievals.

We included a paragraph in Section 3.7 discussing this. We also added Appendix A and Fig. A1, which analyze the CALIOP profiles taken closest to the volcano on 10 April. Because these were night-time lidar measurements, we could only compare them with temperature-based plume heights. The radiometric–lidar height comparison showed strong similarities to the daytime radiometric–geometric height comparison given in Fig. 16.

*L24, regarding the 1979 eruption: Please provide a citation if one exists.*

Added a reference to Fiske and Sigurdsson (1982).

*L60, "dark pixel": I am not aware of this term. Please provide a definition and/or citation.*

We introduce the term "dark pixel temperature" as a synonym for "the minimum 11µm brightness temperature" in the 2nd paragraph of the Introduction. If the IR image is plotted using a dark-to-bright color palette (e.g., black and white), the darkest pixel represents the minimum brightness temperature of the plume. We included a reference to de Michele et al. (2019), who also uses this term. In addition, we provided two more references for the brightness temperature method (Oppenheimer, 1998; Prata and Grant, 2001) per the request of Referee #1.

*L80, Regarding pixel resolution. Is a citation needed?*

Added a reference to Kalluri et al. (2018).

*L86, "Such uncertainty can still be competitive for…": It is unclear what is meant by "competitive" here. What is competing with what?*

Changed "competitive" to "acceptable".

*L99, "As a result,": As a result of what? The actual cold point temperature? How does that translate to "~220K"? In general, any BT will have more than one solution as long as there is a layer warmer than that in the atmospheric column. So, is 220K simply chosen here as a practical value considering eruption column range? Please clarify.*

We replaced "As a result" with "For this profile". We also clarified that the temperature profile in Fig. 4 is plotted for the eruption height range (<24km).

*L102, "…opposite sign, but comparable magnitude…": What is the significance of the comparable magnitudes? Presumably that has no bearing on how meaningful the midpoint plume height is. Perhaps the authors should at this point explain how the midpoint is to be interpreted. If the cloud is opaque, an assigned plume height in between the two points on the T profile is seemingly arbitrary. So, some justification is needed.*

The only observation we try to make here is that for a tropical temperature profile, the midpoint height is near the tropopause, because the tropospheric and stratospheric lapse rates are of similar magnitude but opposite sign. For a temperature profile with a nearly isothermal lower stratosphere, such as at the higher latitude Raikoke volcano, this would not be the case. For a certain plume temperature range, the midpoint height is the best match to the geometric height, as shown in Sect. 4.1. We added a sentence to better clarify the practical 'significance' of the midpoint height.

*L89, "2.2 GOES-16 brightness temperatures": This is just a thought, not a suggestion for the current manuscript…Have the authors considered invoking a radiometric cloud-top topography metric? By that I mean defining a simple cloud with a single, local BT minimum (e.g. Figure 3g) with all surrounding pixels having warmer BT. To first order, this is what one would find when the column is rising up toward the cold-point. When the column enters the tropopause zone, the topography may be expected to get wavier, with multiple local BT minima and maxima. This wavy/complex topography can be distinguished from the "simple" topography and be used as an indicator of greater uncertainty in the BT/z lookup result.*

We thank Reviewer for this valuable suggestion. The second phase of our VolImpact project just started a couple of months ago. We didn't delve deeply into the evolution of the plume-top brightness temperature and its effect on BT-based height estimation in this report, because an entire work package is dedicated to this topic in VolImpact.

We will analyze several recent eruptions (Raikoke, Ulawun, La Soufriere, and HT-HH) and will also use radiative transfer models to understand the plume-top BT. La Soufriere will be our primary case thanks to the availability of the 1-minute MESO data. The proposed radiometric cloud-top topography metric can certainly be tested too.

*L173, 174, "Because the OT can be assumed to exhibit only small downwind advection and thus to lie nearly above the vent,…": Why resort to assumption, when we can look at wind profile (radiosonde or reanalysis data) to assess forcing on the column top tilt? Please expand on this or justify the assumption.*

According to ERA5, there were 16 m/s westerly winds at the tropopause, which could certainly introduce a height error by bending the column. However, the fact that the OT is located along the view azimuth direction indicates small bending for this strong plume. Bending by the wind is anyway better calculated by a plume model such as FPLUME, which accounts for the entire wind profile as well as the exit velocity (weak vs. strong plume); but this would be an overkill. We edited this paragraph to justify our assumption and to clarify that this is only a rough back-of-the-envelope estimate here.

*L181, "This cold bias is likely the consequence of observing a warm subpixel stratospheric target above a colder umbrella spreading at the tropopause, combined with potential thermal disequilibrium due to…": This sentence is long and unclear. We're talking about the coldest pixel. All the other BTs are warmer. What is meant by the cold umbrella? Please reword for clarity.*

We deleted this sentence and reworded the preceding one.

*L221, "The GOES-16 plume-top BT11 shows a cold…": Citation needed for this sentence.*

Added a reference to Setvák et al. (2013).

*L223, 224, "The minimum BT11 of 197.6K is located considerably downwind of the volcano, over an optically thick and opaque part of the umbrella.": How do we know about the optical opacity there? It is not self-evident from the figure. Please explain.*

Fair comment. We did not apply any objective tests to evaluate opacity, therefore, we removed the second half of the sentence.

*Speaking of opacity, have the authors considered citing and applying works such as Inoue (1987), who employed split-window BTD to independently characterize cloud opacity? Since the issue of semi-transparency is a theme here, it may be essential for the authors to demonstrate usage of tested means for evaluating the opacity of selected IR BT pixels.*

As mentioned above, the detailed analysis of the evolution of plume-top BT, including radiative transfer modelling and opacity testing, is deferred to an upcoming study. We will certainly revisit the split-window technique of Inoue (1987).